# Modulatory Effects of the Kuwanon-Rich Fraction from Mulberry Root Bark on the Renin–Angiotensin System

**DOI:** 10.3390/foods13101547

**Published:** 2024-05-16

**Authors:** Ji-Hae Lee, Heon-Woong Kim, So-Ah Kim, Wan-Taek Ju, Seong-Ryul Kim, Hyun-Bok Kim, Ik-Seob Cha, Seong-Wan Kim, Jong-Woo Park, Sang-Kuk Kang

**Affiliations:** 1Department of Agricultural Biology, National Institute of Agricultural Sciences, Rural Development Administration, Wanju 55365, Republic of Koreacha8352@korea.kr (I.-S.C.);; 2Department of Agro-Food Resources, National Institute of Agricultural Sciences, Rural Development Administration, Wanju 55365, Republic of Korea

**Keywords:** mulberry, kuwanon, root bark, renin, angiotensin converting enzyme

## Abstract

In this study, we investigated the anti-hypertensive properties of mulberry products by modulating the renin–angiotensin system (RAS). Comparative analysis showed that the ethyl acetate fractions, particularly from the Cheongil and Daeshim cultivars, contained the highest levels of polyphenols and flavonoids, with concentrations reaching 110 mg gallic acid equivalent (GE)/g and 471 mg catechin equivalent (CE)/g of extract, respectively. The ethyl acetate fraction showed superior angiotensin-converting enzyme (ACE) inhibitory activity, mainly because of the presence of the prenylated flavonoids kuwanon G and H. UPLC/Q-TOF-MS analysis identified kuwanon G and H as the primary active components, which significantly contributed to the pharmacological efficacy of the extract. In vivo testing of mice fed a high-salt diet showed that the ethyl acetate fraction substantially reduced the heart weight and lowered the serum renin and angiotensinogen levels by 34% and 25%, respectively, highlighting its potential to modulate the RAS. These results suggested that the ethyl acetate fraction of mulberry root bark is a promising candidate for the development of natural ACE inhibitors. This finding has significant implications for the management of hypertension through RAS regulation and the promotion of cardiovascular health in the functional food industry.

## 1. Introduction

Hypertension is recognized globally as a chronic disease and a leading cause of premature death, affecting approximately 1.28 billion adults worldwide [1,2]. This is mainly regulated by the renin–angiotensin system (RAS), which plays a central role in regulating blood pressure and electrolyte balance in the body [3]. Renin, released by the kidneys, converts liver-produced angiotensinogen into the decapeptide angiotensin I (AngI). This molecule then undergoes further modification when angiotensin-converting enzyme (ACE), located in the vascular endothelium, cleaves the Phe-His dipeptide from AngI to generate angiotensin II (Ang II) [3]. Ang II functions as a potent vasoconstrictor when it binds to specific receptors known as AT1 and AT2. These receptors facilitate downstream signaling that enhance blood pressure regulation [4]. Activation of the AT1 receptor stimulates phospholipase C (PLC), which subsequently catalyzes the production of inositol-1,2,3-trisphosphate (IP3). IP3 plays an important role by inducing the release of calcium ions (Ca^2+^). Elevated Ca^2+^ levels activate myosin light-chain kinase (MLCK), which in turn phosphorylates myosin, resulting in vasoconstriction and an increase in blood pressure [5]. The RAS is considered a key system in regulating blood pressure; therefore, research on lowering blood pressure through RAS regulation and its active substances has been conducted.

The mulberry fruit (*Morus alba* L.) contains various bioactive substances, including anthocyanins, and it has antioxidant, neuroprotective, anti-atherosclerotic, immunomodulatory, anticancer, hypoglycemic, and lipid-lowering properties. Therefore, it is widely used as a fruit in traditional medicine [6]. Although mulberry leaves have been used as silkworm feed, recent studies have identified specific functional components such as 1-deoxynojirimycin (DNJ) and rutin. These components are recognized for their potential health benefits and are now widely used as food ingredients. DNJ, which is structurally similar to glucose, acts as a competitive inhibitor of glucosidase, thereby inhibiting the increase in postprandial blood glucose levels. Rutin has been reported to activate thermogenic responses, among other physiological activities [7]. Furthermore, mulberry leaves contain γ-aminobutyric acid (GABA), an amino acid that is a major inhibitory neurotransmitter in the central nervous system. GABA is associated with regulating blood pressure by reducing noradrenaline secretion [8,9].

The root bark of mulberry is traditionally used as a tea or medicinal herb and has high academic and industrial value owing to its distinct composition of bioactive compounds. The major constituents include the stilbene series of resveratrol, oxyresveratrol, and mulberroside, and a large number of flavonoid components, such as morusin and kuwanon. According to a study by Kim et al. (2022) [10], these compounds enable mulberry root bark to play a pivotal role in managing circulatory diseases by effectively inhibiting cholesterol synthase enzymes and preventing thrombosis. The beneficial effects are attributed to the synergistic actions of specific mulberry components, notably oxyresveratrol and morusin, which target and mitigate underlying mechanisms of cardiovascular disorders. Mulberry twigs are registered in the Food Code and consumed as tea, representing an underutilized resource. While mulberry trees are predominantly cultivated for silkworm rearing, leading to the periodic pruning and discarding of twigs during the spring and autumn, these by-products hold untapped potential. The exploration of the physiological activities of mulberry root bark and twigs not only offers an opportunity to upcycle these discarded materials but also supports sustainable agricultural practices. The processed form of these resources, typically extracts, could serve as valuable ingredients in the development of health-promoting products. This approach not only enhances the economic value of mulberry trees but also contributes to the advancement of functional food and nutraceutical sectors, making use of every part of the mulberry tree in an environmentally sustainable manner.

This study aimed to identify the efficacy and active substances in mulberry root bark and twigs for chronic circulatory diseases such as hypertension. Fractions of mulberry root bark and twig were prepared, and the activity of the blood pressure-regulating enzyme ACE was measured in each fraction. The functional components of the mulberry fractions were also analyzed, and a comparative analysis of the ACE enzyme activity among the functional components was conducted. Additionally, an evaluation of whether the blood pressure regulation system was organically regulated was performed using a high-salt-diet mouse model. This study’s results are intended to serve as scientific evidence for the development of functional mulberry materials to treat and prevent hypertension.

## 2. Materials and Methods

### 2.1. Sample Preparation

The mulberry root barks of four cultivars (Cheongil, Cheonol, Daeshim, and Gwasang2) and the twigs of two cultivars (Cheongil and Cheonol) were harvested from the National Institute of Agricultural Sciences (Wanju, Korea) in 2021. The roots and twigs were then dried and ground for extraction. Methanol (70%, *v*/*v*; J.T. Barker, Phillipsburg, NJ, USA) was added to 5 g of each dried powder, and extraction was performed by sonication for 30 min. The supernatants were collected and filtered through a PVDF membrane (0.45 μm, Thermo Fisher, Waltham, MA, USA). The extracts were concentrated using nitrogen gas and fractionated. Each fraction was obtained by sequentially adding n-hexane, dichloromethane, ethyl acetate, and n-butanol purchased from J.T. Barker. The fractions were concentrated and lyophilized.

### 2.2. Crude Polyphenol and Flavonoid Quantification

The polyphenol and flavonoid contents of the twigs and roots were analyzed using a colorimetric method. The total polyphenol content was determined by reacting 10 μL of the sample with a 2% Na_2_CO_3_ solution for 3 min at room temperature and then adding 10 μL of a 50% Folin–Ciocalteu solution (Sigma-Aldrich, St. Louis, MO, USA). After a reaction time of 3 min, the absorbance was measured at 750 nm, and the concentration was calculated. To measure the flavonoids, 100 μL of DW and 6 μL of 5% NaNO_2_ (Sigma-Aldrich) were added to the sample (10 μL) and allowed to react for 5 min. Next, 12 μL of AlCl_3_-6H_2_O (Sigma-Aldrich) 10% solution was added, and the reaction was allowed to continue for another 5 min. The reaction was terminated by adding 1 M NaOH (40 μL) and the absorbance was measured at 510 nm. Gallic acid and catechin (Sigma-Aldrich) were used as standards for the quantification of the polyphenols and flavonoids, respectively.

### 2.3. Analysis of the Inhibition Rate of Angiotensin-Converting Enzyme

The enzymatic activity of ACE1 was measured using an assay kit according to the manufacturer’s instructions (Abcam, Ab283407, Cambridge, UK). Briefly, samples were diluted 10-fold with assay buffer, and the reaction was initiated by adding the enzyme solution. After 20 min of reaction, ACE1 substrate was added, and the absorbance of the mixtures was monitored for 1 h at 37 °C. The reading wavelength was 345 nm using a multiplate reader (Multiskan GO, Thermo Fisher Scientific). Standard chemicals were purchased from Sigma-Aldrich (resveratrol and oxyresveratrol) and PhytoLab (Scottsdale, AZ, USA; mulberroside A, kuwanon G, and kuwanon H).

### 2.4. Quantification of Kuwanon G and Kuwanon H in Extracts of Mulberry Root Bark

The concentrated ethyl acetate solution was redissolved in 70% methanol. The test solution was prepared through a polyvinylidene fluoride (PVDF, 0.22 μm) membrane filter (Thermo Fisher). The samples were injected into an ultra-performance liquid chromatography-diode array detector system (UPLC-DAD) coupled with quadrupole time-of-flight (QToF) mass spectrometry (SCIEX X500R, SCIEX Co., Framingham, MA, USA). The main and pre-columns were configured as CORTECS UPLC T3 (2.1 × 150 mm, 1.6 μm, Waters Co., Milford, MA, USA) and CORTECS UPLC Vanguard T3 (2.1 × 50 mm, 1.6 μm, Waters Co.), respectively. The chromatographic conditions employed were as follows: flow rate (0.3 mL/min), column temperature (30 °C), injection volume (1 μL), mobile phases (A, 0.5% formic acid in water; B, 0.5% formic acid in acetonitrile), gradient profile (initial 5% B; 20 min, 25% B; 25 min, 50% B; 30–32 min, 90%, 35–40 min, 5% B). The profile of the gradient was as follows: 5% B for 20 min, 25% B for 25 min, 50% B for 30–32 min, 90% B for 35–40 min, and 5% B. The ultraviolet spectra were scanned in the wavelength range of 210–400 nm (representative wavelengths: 264 nm for kuwanon G and kuwanon H). The mass spectra were simultaneously measured using an electrospray ionization (+ESI) source in positive ionization mode in the range of *m*/*z* 100–1200. The following parameters were used: ion source gas, 50 psi; curtain gas, 30 psi; ion source temperature, 450 °C; declustering potential (DP), 80 V; collision energy (CE), 25 ± 10 V; spray voltage, 5500 V. The DP was set at 80 V, the CE at 25 ± 10 V, and the spray voltage at 5500 V. The kuwanon G and kuwanon H contents (PhytoLab GmbH & Co., Vestenbergsgreuth, Germany) were externally measured using calibration curve data based on the peak area, with the corresponding concentrations serving as the basis for this measurement.

### 2.5. Animal Study

Hypertension was induced in C57BL/6J mice (female, 4 weeks old; Central Lab Animal, Seoul, Korea) using an 8% high-salt diet (HSD, Research Diets, New Brunswick, NJ, USA) [11]. After 1 month of HSD administration, the mice were randomly divided into three groups (*n* = 8). The three groups were fed an HSD diet or fed an HSD with 70% methanol extract (root bark extract, RBE) or ethyl acetate fraction (root bark fraction, RBF). The test extract and fractions were combined in the HSD diet at a consumption rate of 100 mg/kg/day. The test diets were fed for 3 months, and weight gain was monitored. At the end of the feeding period, the mice were sacrificed, and liver, kidney, heart, and blood samples were collected for analysis. The animals were maintained at 23 ± 1 °C and 56% relative humidity. The animal experiments were approved by the Animal Experiment Ethics Committee of the National Institute of Agricultural Sciences (NAAS-2023010).

### 2.6. Mouse Serum Analysis

Blood samples were collected from the mouse heart after sacrifice by injecting 2,2,2-tribromoethanol (Sigma-Aldrich). The serum was separated by centrifugation at 2000 rpm for 10 min. The renin and angiotensinogen concentrations were measured using an enzyme-linked immunosorbent assay (ELISA) according to the manufacturer’s instructions. For the serum renin analysis kit (Abcam, ab193728), the serum was diluted 10-fold, and the reaction was initiated with a renin detection antibody for 1 h. Horseradish peroxidase (HRP) solution was added and incubated for another 45 min. The tetramethylbenzidine (TMB) substrate was added and allowed to stand for 30 min, and the samples were stopped by adding stop solution. The absorbance was measured at 450 nm. A product from Abcam (ab245718) was used for the angiotensinogen analysis. The serum samples were diluted 2000 times with buffer and incubated with an antibody cocktail for 1 h. The TMB solution was added, and the reaction was stopped with a stop solution after 10 min. The reactants were detected at a wavelength of 450 nm.

### 2.7. Statistical Analysis

A one-way analysis of variance was conducted using SPSS Statistics 23 (IBM, Armonk, NY, USA) to determine the significance of the means. Duncan’s multiple-range test was used to assess the statistical significance of the results at a 0.05 level of significance (*p* < 0.05). The 95% confidence intervals (CI) of the mean values were calculated (Table 1 and Appendix A).

## 3. Results and Discussion

### 3.1. Comparison of Crude Polyphenol and Flavonoid Contents of Mulberry Root Bark and Twig Extracts

The total polyphenol and flavonoid contents of mulberry twigs and root bark were compared according to the fraction and cultivar (Figure 1). The highest amounts of polyphenol and flavonoid were detected in the ethyl acetate fraction of the root bark from the Cheongil cultivar (110 mg GE/g of extract) and the ethyl acetate fraction of the root bark from Daeshim (471 mg CE/g of extract), respectively. The dichloromethane and ethyl acetate fractions were more effective in extracting flavonoids than the butanol- and water-soluble fractions. The root bark contained higher amounts of polyphenols and flavonoids than the twigs. The polyphenol and flavonoid contents per gram increased during the fractionation process compared to those in the methanol extraction, indicating that the fractionation process concentrates useful substances. 

Mulberry twigs and root barks are commonly used in traditional medicine in East Asian countries. These plant parts contain specific bioactive substances such as alkaloids, stilbenes, and prenylated flavonoids. In addition to various secondary metabolites, mulberry twigs and root bark have been shown to possess antioxidant, anti-diabetic, anti-inflammatory, and anticancer properties. The antioxidant and anti-inflammatory capacity of mulberry helps to reduce oxidative stress and the risk of chronic inflammatory diseases. The anticancer potential is particularly recognized for its ability to suppress tumor growth and induce apoptosis in cancer cells [12,13]. However, it is important to note that the activity of these substances can vary depending on the extraction conditions. The extraction conditions, such as the solvent type and temperature, play a crucial role in maximizing their therapeutic benefits. According to recent studies, optimal extraction parameters are essential to ensure the highest yield and effectiveness of bioactive substances like alkaloids, stilbenes, and prenylated flavonoids. These conditions significantly affect the structural characteristics and antioxidant activities of the extracts, highlighting the need for precise control over the extraction process to harness the full potential of mulberry components [14]. Since different compounds require different elution conditions, it is necessary to identify the extraction conditions that maximize the bioactivity of each compound and to identify the active compound responsible for each health-promoting effect. Therefore, further studies were conducted to identify the hypotensive-active fractions and compounds. 

### 3.2. Inhibitory Effect of Mulberry Twig and Root Bark Extract on ACE Activity

The ACE inhibitory effects of the fraction and part of the mulberry were compared (Figure 2). ACE was most effectively inhibited by the ethyl acetate and dichloromethane fractions. At a 10 μg/mL concentration, the methanol extract, dichloromethane, ethyl acetate, butanol, and DW fractions showed 23, 81, 95, 6, and 0% inhibitory effects, receptively. In the comparison of mulberry parts, the inhibition rate of root bark was higher at 92–102% and twig at 42–62%. The inhibition rates of the roots of Cheongol and Cheongil were significantly higher than those of Gwasang2 (*p* < 0.05).

Hypertension is characterized by a chronic elevation of blood pressure [15,16,17]. Various medications are currently available, including diuretics, calcium channel blockers, α-blockers, β-blockers, vasodilators, central sympatholytics, and ACE inhibitors. Each class of drug has unique mechanisms of action and indications. For example, diuretics facilitate the removal of excess fluid from the body and are often used as the first-line treatment for hypertension. Calcium channel blockers lower blood pressure by inhibiting the flow of calcium into cardiac and smooth muscle cells, resulting in vasodilation. Moreover, α-blockers and β-blockers target specific receptors to reduce nerve impulses associated with vasoconstriction and heart rate. Vasodilators relax vascular smooth muscle cells directly. Central sympatholytics lower blood pressure by inhibiting signals from the brain that cause blood vessels to constrict [18,19,20]. ACE inhibitors are characterized by lowering blood pressure by specifically blocking the conversion of angiotensin I to angiotensin II, preventing vasoconstriction and water retention [21,22,23]. Subsequently, these treatments are pivotal not only for managing blood pressure but also for preventing the long-term cardiovascular complications associated with hypertension. A meta-analysis encompassing 20 trials on hypertension demonstrated that ACE inhibitors correlate with a 10% decrease in the all-cause mortality risk and a 12% reduction in the cardiovascular mortality risk [24]. Therefore, the ACE inhibition activity of the mulberry root bark fraction showed efficacy as a blood pressure regulator. The single substance experiments were conducted to identify the specific active substance.

### 3.3. Comparison of Single Compounds in ACE Inhibition

The ACE enzyme inhibition effects of single compounds derived from mulberries were compared (Figure 3). Stilbene and prenylated flavonoids commonly observed in mulberries were selected. In an experiment treated with a concentration of 10 μg/mL, stilbene-based substances had no enzyme inhibitory effect. In contrast, kuwanon G and H, which are prenylated flavonoids, exhibited inhibitory effects (8–19%). Comparison of kuwanon G and H showed concentration-dependent inhibition, and the effect of kuwanon H was 2.2-fold higher than that of kuwanon G at 100 μg/mL.

Prenylated flavonoids are a distinct class of natural compounds, notable for their flavonoid structures, which are modified with additional prenyl or geranyl side chains [25]. The molecular properties of these prenyl groups enhance the lipophilicity of flavonoids, increase the permeability of cellular phospholipid membranes, and improve the interactions with target proteins. This enhances the biological activity of flavonoids, increasing their potential therapeutic effects and interactions within biological systems [26]. Prenylated flavonoids have been extensively studied and demonstrated diverse pharmacological properties, including potent antibacterial, antiviral, anti-inflammatory and antioxidant activities. These effects highlight their potential as therapeutic agents in the treatment and management of a variety of conditions resulting from microbial infection, inflammation and oxidative stress [27,28,29,30]. In addition, bioactivity varies depending on the position and number of the prenyl groups. These modifications can significantly influence their molecular properties, including their solubility and how effectively they can interact with biological targets. The specific position and length of the prenyl attachment may affect the stability of flavonoids and their ability to scavenge free radicals, thus impacting the antioxidant efficacy. Such structural variations are crucial for optimizing the medicinal potential of prenylated flavonoids [31,32]. Accordingly, kuwanon G and H had one and two prenyl groups, respectively, and showed different ACE enzyme inhibitory effects in this respect. It is suggested that the position and number of the prenyl groups influence the ACE enzyme inhibition.

### 3.4. Quantification of Kuwanon G and Kuwanon H in Root Bark and Twig Fraction with ACE Inhibitory Effect

The quantification of the phenolic compounds in the ethyl acetate fraction, which had the highest ACE enzyme inhibition effect, was conducted using UPLC-DAD-QToF/MS. Kuwanon G and H, which are prenylated flavonoids, were confirmed as the predominant compounds in the ethyl acetate fraction of the root bark (Figure 4) [33,34]. The contents of these compounds (mg/g of extract) were distributed differently according to the cultivar type (Cheongol, Cheongil, Daeshim, and Gwasang2). Cheongol had the highest content of prenylated flavonoids (255.5 mg), which were composed of kuwanon G (173.3 mg) and kuwanon H (82.2 mg), in relation to ACE inhibition (Table 1). 

Secondary metabolites are often species- and cultivar-dependent, similar to observations in mulberry. Previous studies have analyzed the fruits of 12 Korean mulberry cultivars, revealing differences not only in the amount of compounds but also in the presence or absence of certain components among the cultivars [35]. This study compared the ethyl acetate fractions of four common Korean mulberry cultivars. Cheongol and Cheongil are mainly grown for sericulture, while Daeshim and Gwasang2 are grown for fruit harvest. The results showed that the cultivars used for sericulture, such as Cheongol and Cheongil, had higher total kuwanon contents than those grown for fruit harvest. This suggests that the selection of cultivars for their higher bioactive components could be crucial for the development of more effective extracts, providing a guideline for cultivar selection in future studies to exploit the best therapeutic potential of mulberry. The results of this study showed that Cheongol had the highest content of kuwanon G and H. However, since the ingredient content varies depending on the harvest time of the sample, age of the tree, climate, cultivation area, etc., future research is needed on the optimized conditions for harvest.

### 3.5. Effect of Root Bark Extract and Ethyl Acetate Fraction on High-Salt-Diet-Fed Mice

The hypertensive effects of the 70% methanol extract (RBE) and the ethyl acetate fraction (RBF) of the root bark were compared in an HSD mouse model (Figure 5). Three months after administration, there were no changes in the body, liver, or kidney weights (*p* > 0.05). In contrast, the heart weight, which is related to blood pressure, was significantly reduced by 8% in the RBF group (*p* < 0.05). In addition, the serum concentrations of renin and angiotensinogen were reduced by 34% and 25%, respectively, in the RBF group (*p* < 0.05). The effect of the RBF was consistently more pronounced than that of the RBE.

High blood pressure is known to cause structural remodeling of the heart, often resulting in increased heart weight and the development of conditions such as left ventricular hypertrophy. This remodeling can have a serious impact on heart function and, if not managed effectively, may make an individual more susceptible to developing cardiovascular complications. Hypertensive heart disease, characterized by these structural changes, is an important indicator of increased cardiovascular risk, highlighting the need for timely and effective blood pressure management [36,37,38]. RBF, in a group consuming fractions with high kuwanon content, showed a significant decrease in heart mass compared to an HSD, confirming that it has a protective effect on cardiac structural changes due to blood pressure. The RAS plays an important role in the regulation of blood pressure [39,40]. Renin, secreted by the kidneys, starts a chain reaction by converting angiotensinogen to angiotensin I in the liver. This peptide is converted to the potent vasoconstrictor angiotensin II by ACE, which is primarily localized in the lungs and endothelial cells. Angiotensin II primarily acts through the angiotensin type 1 receptor (AT1R) to induce vasoconstriction and sodium retention, thereby raising the blood pressure. This pathway, which is key to blood pressure regulation, has been extensively validated and is known to be a key therapeutic mechanism for the treatment of hypertension [41]. Fractions enriched in kuwanon and extract effectively lowered the blood renin and AGT, respectively. In addition, the decrease in RBF was greater than that in RBE. The efficacy of RBE in inhibiting ACE activity is shown in Figure 2, and the ACE activity inhibition effect of the main constituents was also verified in Figure 3. The regulatory effect of RAS through animal experiments, together with its ACE enzyme inhibitory activity, further supported the efficacy of mulberry root bark as an antihypertensive agent.

## 4. Conclusions

This study investigated the antihypertensive effects of mulberry root bark, focusing on the RAS-inhibitory potential of its ethyl acetate fraction. Our study revealed that this specific fraction, which was significantly enriched in kuwanon G- and H-prenylated flavonoids, exhibited the most potent ACE inhibitory effects. Detailed component analysis using UPLC/Q-ToF-MS identified kuwanon G and H as the dominant bioactive compounds responsible for the observed pharmacological activity. In vivo experiments in a high-salt-diet-fed mouse model showed that the administration of the ethyl acetate fraction resulted in a significant decrease in the heart weight and serum concentrations of renin and angiotensinogen, key regulators of RAS, confirming its efficacy in blood pressure regulation. These findings suggest that the ethyl acetate fraction of mulberry root bark, with its high content of kuwanon, is a valuable natural resource for the development of functional food ingredients or dietary supplements for the prevention and treatment of hypertension, offering a promising approach to mitigating cardiovascular disease through dietary intervention. There were limitations to the current study in terms of the toxicity assessment and hypertensive effect through clinical trials, which should be considered in further studies to evaluate the commercial use of mulberry root bark.

## Figures and Tables

**Figure 1 foods-13-01547-f001:**
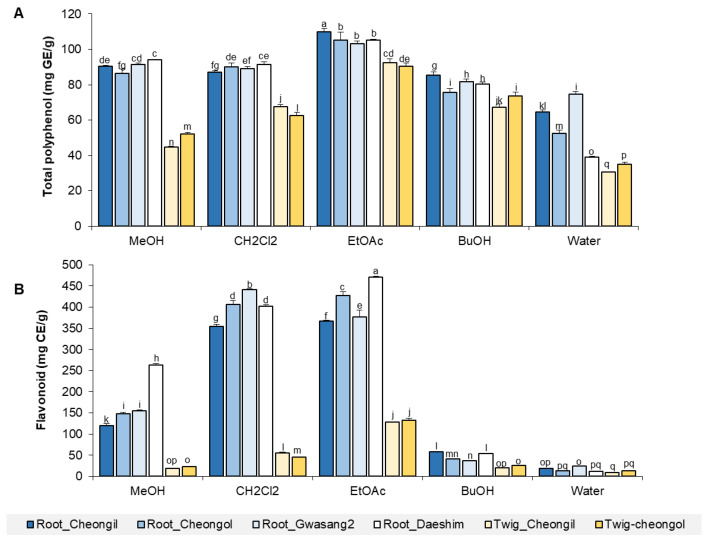
Comparison of total polyphenol and flavonoid contents of mulberry root bark and twig extracts across different cultivars. Total polyphenol (**A**) and flavonoid (**B**) contents were measured in various fractions of mulberry root bark and twig extracts from different cultivars (Cheongil, Cheongol, etc.). Bars represent the concentration of polyphenols and flavonoids in mg GE/g or mg CE/g of extract, with statistical significance denoted by different letters above the bars (*p* < 0.05), *n* = 3. The CI value is shown in Appendix A.

**Figure 2 foods-13-01547-f002:**
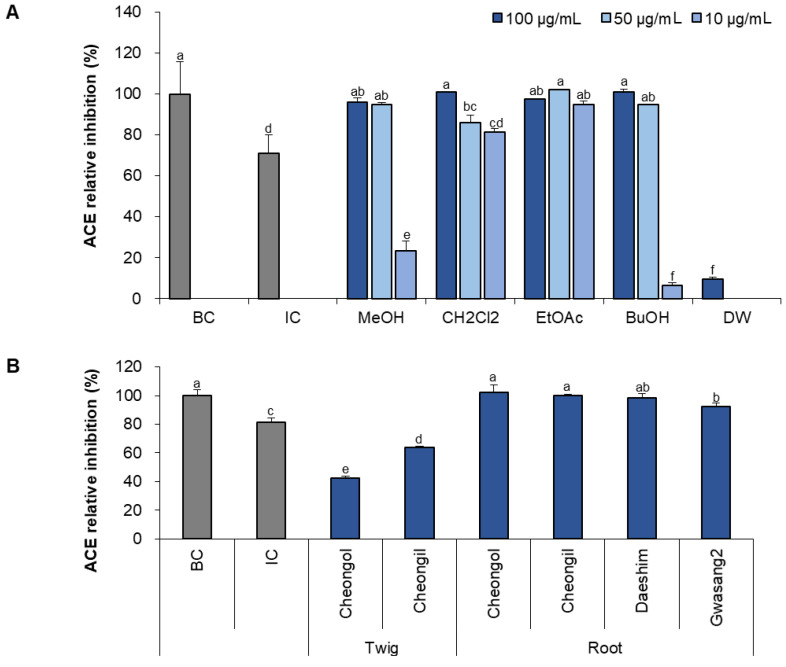
ACE inhibitory effects of mulberry root bark and twig extract fractions. The ACE inhibitory rate (%) of different root bark fractions at 10–100 μg/mL (**A**) and cultivars at 100 μg/mL (**B**) were compared. The efficacy of the inhibition is categorized by the fraction or the source (root vs. twig). Abbreviations are background control (BC) and inhibitor control (IC). Statistical significance is denoted by different letters above the bars (*p* < 0.05), *n* = 3.

**Figure 3 foods-13-01547-f003:**
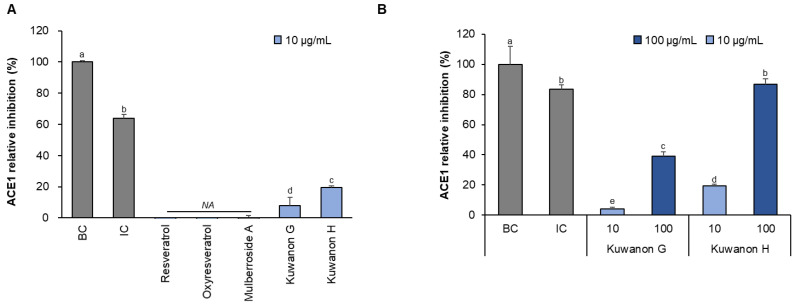
Comparative ACE inhibition by single compounds derived from mulberry. The ACE inhibitory effects of single compounds at 10 μg/mL (**A**) and kuwanons according to the density (10–100 μg/mL) (**B**) were compared. Abbreviations ae background control (BC), inhibitor control (IC), and not applicable (NA). The differences between groups were compared and the significance levels marked accordingly using different letters (*p* < 0.05), *n* = 3.

**Figure 4 foods-13-01547-f004:**
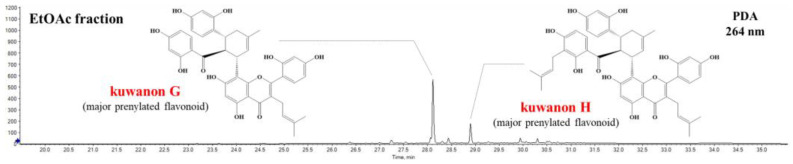
UPLC chromatogram of ethyl acetate fraction from mulberry root bark. Chromatogram of the ethyl acetate fraction from mulberry root bark highlighting the major peaks corresponding to prenylated flavonoids, specifically kuwanon G and H. The chromatogram was recorded at 264 nm, demonstrating the prominence of these compounds within the fraction.

**Figure 5 foods-13-01547-f005:**
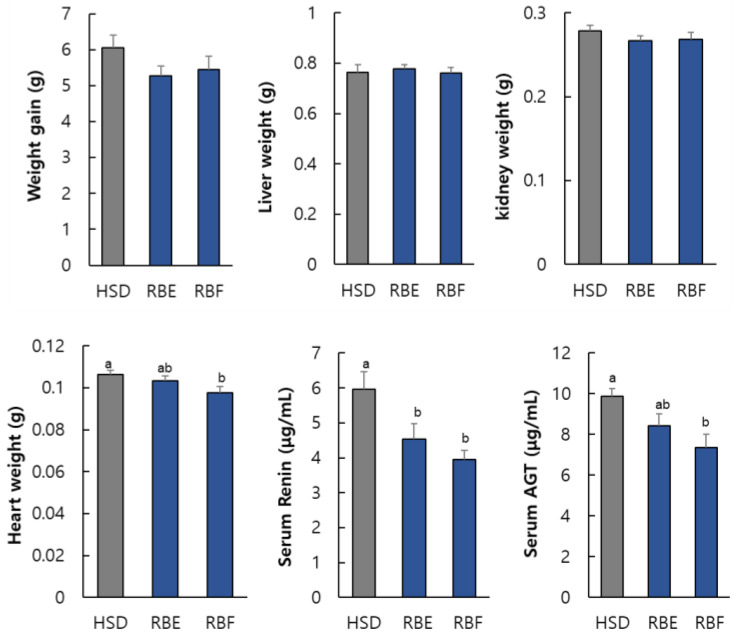
Effects of mulberry root bark extract on high-salt-diet-induced hypertension in mice. The effects of 70% methanol extract (RBE) and ethyl acetate fraction (RBF) from mulberry root bark on heart weight and serum renin and angiotensinogen levels in mice fed a high-salt diet were analyzed. Results are categorized into the control, RBE, and RBF groups, with statistical significance shown using different letters (*p* < 0.05), *n* = 8.

**Table 1 foods-13-01547-t001:** Quantitative analysis of major flavonoids in ethyl acetate fractions by mulberry cultivar.

Peak NO.	Flavonoid	Cheongol	Cheongil	Daeshim	Gwasang2
1	Kuwanon G	173.3 ± 8.9	140.4 ± 5.1	16.2 ± 1.5	165.4 ± 7.3
	(95% CI)	(151.17–195.52)	(127.82–152.94)	(12.59–19.86)	(147.35–183.39)
2	Kuwanon H	82.2 ± 4.2	80.5 ± 3.9	12.9 ± 1.4	55.6 ± 2.7
	(95% CI)	(71.87–92.62)	(70.78–90.23)	(9.48–16.40)	(48.77–62.35)

Units are mg/g extract; dry matter. CI indicates confidence interval. Each value is calculated as mean ± SD, *n* = 3.

## Data Availability

The original contributions presented in the study are included in the article/Appendix A, further inquiries can be directed to the corresponding author.

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
