# Peer review of "Modulatory Effects of the Kuwanon-Rich Fraction from Mulberry Root Bark on the Renin–Angiotensin System"

_foods, 2024, doi:10.3390/foods13101547_

Round 1
Reviewer 1 Report
Comments and Suggestions for Authors
To authors
The comment on the manuscript is in the PDF attachment file herewith; please check it out.
Best regards,

Reviewer 2 Report
Comments and Suggestions for Authors
Title:
“Modulatory effects of kuwanon rich fraction from mulberry root bark on the renin-angiotensin system (RAS)”
Review
The manuscript studies the effect of kuwanon from mulberry root bark on the renin-angiotensin system of mice. The objective of this review is to try to improve the manuscript. Some indications can be addressed.
1. The statistical analysis could be more detail, including the sample size necessary to probe the authors’ hypothesis considering a sufficient power, considering the effect of kuwanon G and H on mice. In addition, how comparisons of different cultivars were performed.
2. The number of cultivars and mice involved in the experiments may be indicated for better understanding the results.
3. The authors only use p-values to show results. Some tables with results may be convenient, considering that p-values are dependent on sample size. In addition, a confidence interval of results may be clarified.
4. The authors only use p-values to show results. Some tables with results may be convenient, considering that p-values are dependent on sample size. In addition, a confidence interval of results may be clarified.
5. The authors mention three groups of mice to compare the results. However, there was no indication how the groups were made, and there are not Tables to show results.
6. No limitations of this study are indicated. However, a potential toxicity from the use of kuwanon may be mentioned.
Round 2
Reviewer 2 Report
Comments and Suggestions for Authors
Explain the acronym BC and IC in several Tables.
Author Response
Response to reviewers
We are submitting the revised manuscript in accordance with the comment provided by the reviewer.
Reviewer
- Explain the acronym BC and IC in several Tables
> Answer: The legend in Figure 2-3 have has been revised based on a reviewer’s comment.
